# Current Overview on Therapeutic Potential of Vitamin D in Inflammatory Lung Diseases

**DOI:** 10.3390/biomedicines9121843

**Published:** 2021-12-06

**Authors:** Muhammad Afzal, Imran Kazmi, Fahad A. Al-Abbasi, Sultan Alshehri, Mohammed M. Ghoneim, Syed Sarim Imam, Muhammad Shahid Nadeem, Maryam Hassan Al-Zahrani, Sami I. Alzarea, Ali Alquraini

**Affiliations:** 1Department of Pharmacology, College of Pharmacy, Jouf University, Sakaka 72341, Saudi Arabia; afzalgufran@ju.edu.sa (M.A.); samisz@ju.edu.sa (S.I.A.); 2Department of Biochemistry, Faculty of Science, King Abdulaziz University, Jeddah 21589, Saudi Arabia; fabbasi@kau.edu.sa (F.A.A.-A.); mhalim@kau.edu.sa (M.S.N.); mhsalzahrani@kau.sdu.sa (M.H.A.-Z.); 3Department of Pharmaceutics, College of Pharmacy, King Saud University, Riyadh 11451, Saudi Arabia; salshehri1@ksu.edu.sa (S.A.); simam@ksu.edu.sa (S.S.I.); 4Department of Pharmacy Practice, College of Pharmacy, AlMaarefa University, Ad Diriyah 13713, Saudi Arabia; mghoneim@mcst.edu.sa; 5Department of Pharmaceutical Chemistry, Faculty of Clinical Pharmacy, Al Baha University, Al Baha 65779, Saudi Arabia; aalquraini@bu.edu.sa

**Keywords:** inflammatory lung disorders, cholecalciferol, mechanism, metabolic pathways, treatment

## Abstract

Inflammatory lung disorders (ILDs) are one of the world’s major reasons for fatalities and sickness, impacting millions of individuals of all ages and constituting a severe and pervasive health hazard. Asthma, lung cancer, bronchiectasis, pulmonary fibrosis acute respiratory distress syndrome, and COPD all include inflammation as a significant component. Microbe invasions, as well as the damage and even death of host cells, can cause and sustain inflammation. To counteract the negative consequences of irritants, the airways are equipped with cellular and host defense immunological systems that block the cellular entrance of these irritants or eliminate them from airway regions by triggering the immune system. Failure to activate the host defense system will trigger chronic inflammatory cataracts, leading to permanent lung damage. This damage makes the lungs more susceptible to various respiratory diseases. There are certain restrictions of the available therapy for lung illnesses. Vitamins are nutritional molecules that are required for optimal health but are not produced by the human body. Cholecalciferol (Vitamin D) is classified as a vitamin, although it is a hormone. Vitamin D is thought to perform a function in bone and calcium homeostasis. Recent research has found that vitamin D can perform a variety of cellular processes, including cellular proliferation; differentiation; wound repair; healing; and regulatory systems, such as the immune response, immunological, and inflammation. The actions of vitamin D on inflammatory cells are dissected in this review, as well as their clinical significance in respiratory illnesses.

## 1. Introduction

The majority of the world’s population is affected by pulmonary system inflammation, which is both a serious public health issue and a substantial financial burden. Approximately 510 million individuals are thought to be affected by these diseases globally. Inflammatory lung disorders (ILDs) are one of the world’s major reasons for fatalities and sickness, impacting millions of individuals of all ages and constituting a severe and pervasive health hazard. During the onset and progression of many lung diseases, structural aberrations in several sections of the lung, including the nasal passages and airways, are evident. In both small and large airways, this comprises airway remodeling, parenchymal fibrosis, and epithelial lesions. Additionally, cell proliferation, excessive mucus formation, enhanced collagen accumulation, and the stimulation of molecular pathways such as oxidative stress and inflammation all contribute to the loss of elasticity. Asthma, lung cancer, bronchiectasis, pulmonary fibrosis acute respiratory distress syndrome, and COPD all include inflammation as a significant component [1,2]. Microbe invasions, as well as the damage and even death of host cells, can cause and sustain inflammation. Tobacco smoke; professional sensitizers (coal dust asbestos and silica); ecological toxins both exterior (particulate matter and exhausts of diesel) and interior (second- and third-hand smoke and lead particles); biofuel fumes (charcoal, dung, and wood-burning smoke); irritants (cockroach allergens and house dust mites); and microorganisms (viruses and bacteria), as well as malnutrition and a deficiency of vitamins, can all trigger ILDs. To counteract the negative consequences of irritants, the airways are equipped with cellular and host defense immunological systems that block the cellular entrance of these irritants or eliminate them from airway regions by triggering the immune system. Failure to activate the host defense system will trigger chronic inflammatory cataracts, leading to permanent lung damage. This damage makes the lungs more susceptible to various respiratory diseases [3,4].

Bronchodilators and anti-inflammatory drugs are commonly used in the current therapeutic strategy for pulmonary disorders [5]. The phosphodiesterase-4 inhibitors, methylxanthines, orally as well as an inhaled corticosteroid, and monoclonal antibodies are the anti-inflammatory agents used in the management of ILDs [6]. The muscarinic antagonists and β2 agonists are two types of bronchodilators used in the management of pulmonary disorders [7]. There are certain restrictions to using steroids and bronchodilators. Prolonged use of anti-inflammatory steroids has been linked to glaucoma, the development of cataracts and progression of diabetes, elevated body mass, and GIT hemorrhaging, among the other side effects. Furthermore, patients with severe asthma and COPD may develop steroid resistance [8]. As a result, using steroids as anti-inflammatory medicines in certain illnesses may not be a smart idea. Likewise, the long-term use of bronchodilators has been linked to complications such as convulsions, elevated heart beats, and osteoporosis. Additionally, the usage of methylxanthines is prohibited due to their limited therapeutic index and inability to be utilized long-term [9]. As a result, it is critical to find novel therapeutic agents that are regarded as safe, have few or no adverse effects, and have a broad therapeutic index to successfully treat patients and enhance their overall health [10].

Vitamins are nutritional molecules that are required for optimal health but are not produced by the human body. Cholecalciferol (Vitamin D) is classified as a vitamin, although it is a hormone. Vitamin D is thought to perform a function in bone and calcium homeostasis. Recent research has found that vitamin D can perform a variety of cellular processes, including cellular proliferation; differentiation; wound repair; healing; and regulatory systems, such as the immune response, immunological, and inflammation [11,12]. The actions of vitamin D on inflammatory cells are dissected in this review, as well as their clinical significance in respiratory illnesses.

## 2. Method of Literature Search

We conducted a literature search using thesaurus terms and keywords in the PubMed, Embase, Science direct, Scopus, and Springer Library databases from database inception to 26 August 2020; the following search terms were used: vitamin D, asthma, lung cancer, bronchiectasis, pulmonary fibrosis, acute respiratory distress syndrome, and COPD. There were no language limitations. If data were not accessible, we contacted the authors of the papers. Additionally, we carefully reviewed the included articles’ references for the most recent reviews. We began with a preliminary screening of the titles and abstracts, followed by a full-text review. Exclusions included case reports, case series, duplicate reports, comments, and author reactions.

## 3. The Metabolism of Vitamin D

The prohormone vitamin D must be converted to physiologically active molecules that link to nuclear receptors to regulate a variety of biological activities. The conventional and alternative vitamin D metabolic routes, as well as vitamin D metabolism’s hormonal control, are discussed in this section (Figure 1) [13,14].

## 4. Conventional Metabolic Pathway

Cholecalciferol: vitamin D_3_ and ergocalciferol: vitamin D_2_ are both vitamin D isoforms. In plant fungi and yeasts, in the presence of UV light, ergosterol is converted into vitamin D_2_, which is consumed through a diet rich in plant-based foods like mushrooms. UV light in the epidermis of human skin converts 7-dehydrocholesterol to vitamin D_3_, which may also be found in animal sources like cod liver oil. Vitamin D from the food or in the epidermis of the skin attaches to a protein that binds to vitamin D (VDBP) in the bloodstream and is then transported to the hepatic system. Vitamin D 25-hydroxylase (CYP27A1 and CYP2R1) in the hepatic system converts vitamin D to the calcidiol that is 25(OH)D, which is the most common circulating type of vitamin D in the blood [15,16]. Calcidiol is converted to 1,25-calcitriol in the proximal tubule of the kidney by the hydroxylase of 25(OH)D1 that is a physiologically active type of vitamin D. After entering into the bloodstream and then comparing it to VDBP, calcitriol is transported towards target organs, including the kidneys, gut, and bone, where vitamin D is expected to improve the calcium and phosphate uptake, mobilization, and reabsorption. Since being generated, 24-hydroxylase of 25(OH)D, as that is the main calcitriol downregulating enzyme catalyzing hydroxylation at C-23 and C-24 of both calcidiol and calcitriol, closely regulates their levels. Calcitriol binds to VDRs in target tissues, triggering the nongenomic and genetic control of downwind signaling pathways, also with a wide range of biological activities [17]. Calcitriol interacts with cytosolic VDR in the genomic route, causing it to be phosphorylated, heterodimerized with the retinoid-X receptor (RXR), and then translocated to the nucleus. The composite of calcitriol–VDR–RXR attaches at the promoter region of the vitamin D response element and its target genes and activates transcriptional coactivators to control target gene RNA production and, therefore, a range of activities, including calcium and phosphate metabolism. In hepatic stellate cells, the autophagy adaptor protein p62/SQSTM1 has been shown to perform an important function in heterodimerization and transportation of the RXR–VDR composite to target genes by effectively connecting to RXR, as well as VDR [18]. Calcitriol attaches to VDR, which is bonded to the membrane, also recognized as the 1,25D membrane-associated rapid-response steroid-binding protein, in the nongenomic pathway, and this interplay causes slight alterations in the signaling of the cell, such as Ca^2+^ and MAPK signaling via direct peptide–peptide interactions with signaling molecules, a present intracellularly associated with certain phenotypic functions [19].

## 5. Alternative Metabolic Pathway

A new vitamin D metabolism route involving CYP11A1 has just been discovered. Due to the discovery of CYP11A1 expression in GIT and the epidermis of the skin, it has come out as a novel metabolizing enzyme of calcitriol. 17(OH)D, 22(OH)D, and 20(OH)D are the most common vitamin D metabolites produced by CYP11A1. CYP11A1 hydroxylates these to produce 17,20,23 (OH)3D, 17,20(OH)2D, 20,22(OH)2D, and 20,23(OH)2D [20]. The anti-inflammation effects, as well as differentiation and antiproliferation of the CYP11A1 metabolites, are equivalent to or greater than those of calcitriol in the epidermis of the skin. Importantly, CYP11A1 metabolites have been found to act as biased agonists of VDR [21].

## 6. Vitamin D Metabolism and Hormonal Regulation

Calcitriol has a unique negative feedback process that closely controls the metabolism of vitamin D. The inactivating enzyme of vitamin D, CYP24A1, is one of the calcitriol–VDR–RXR complex’s strongest transcriptional targets. Around 250-bp and 150-bp upstream of the transcriptional initiation codon in the gene promoter of CYP24A1, two VDREs cause a significant stimulation of CYP24A1 by vitamin D, which could also promote CYP24A1 expression by attracting RNA polymerase II and histone H4 acetyl transferases to a region 50–70 kb downstream of the CYP24A1 gene. As a result, in the kidney calcitriol-driven CYP24A1, its expression may closely control the amounts of calcitriol and calcidiol [22,23]. Furthermore, calcitriol suppresses CYP27B1 production in the kidney by a complicated process, including epigenetic changes to the promoter region. Vitamin D metabolism is controlled by two hormones, FGF-23 and PTH, which, together, perform significant functions in regulating calcium and phosphate homeostasis, in addition to the negative feedback controlled by calcitriol. The parathyroid gland secretes PTH in accordance with a reduced blood Ca^+2^ quantity, which is traced by receptors of Ca^+2^-sensing located on parathyroid cells. PTH increases calcitriol synthesis by stimulating renal CYP27B1 expression through methods such as enhanced regulation of the nuclear orphan receptor, the transcription dependent on NR4A2 or enhanced cAMP-dependent transcription [24]. While increasing the calcitriol can cause its breakdown by boosting CYP24A1 expression, PTH can maintain the calcitriol levels by causing CYP24A1 mRNA destruction in the kidney via the stimulation of the cAMP–PKA pathway. As a negative feedback mechanism, the elevated calcium levels caused by persistent calcitriol production can inhibit PTH production by interacting with CaSRs in the parathyroid gland. In reaction to elevated plasma phosphate and calcitriol levels, osteocytes and osteoblasts release FGF-23 [25]. Via interacting with FGF receptor–Klotho complexes in the plasma membrane, FGF-23 promotes phosphate excretion by suppressing sodium–phosphate cotransporter 2 expression, which is found at the apical membranes of PCT. FGF-23 also lowers the blood calcitriol levels by blocking the CYP27B1 expression while increasing the CYP24A1 expression in the renal gland; however, the mechanisms are unknown (Figure 2) [26].

## 7. Vitamin D Consumption and Status

The majority of individuals in the United States take less vitamin D than is advised. According to a 2015 to 2016 National Health and Nutrition Examination Survey (NHANES) analysis, the average daily vitamin D intake from foods and beverages was 5.1 micrograms (204 international units) for men, 4.2 micrograms (168 international units) for women, and 4.9 micrograms (196 international units) for children aged 2–19 years. Indeed, according to the 2013–2016 NHANES data, 92% of males, more than 97% of women, and 94% of persons aged 1 year and older consumed less than the EAR of 10 mcg (400 IU) of vitamin D through foods and drinks [27]. Certain individuals had supplements with extremely high levels of vitamin D. Between 2013 and 2014, an estimated 3.2 percent of the adult population in the United States consumed vitamin D supplements containing 100 mcg (4000 IU) or more of vitamin D. Based on the vitamin D consumption through foods, drinks, and even dietary supplements, one may predict a sizable fraction of the US population to be vitamin D-deficient [28]. Comparing vitamin D intakes to the serum 25(OH)D levels, on the other hand, is difficult. One explanation is that sun exposure influences the vitamin D status, and hence, the blood 25(OH)D levels are frequently greater than would be anticipated just from vitamin D food intakes [29].

## 8. Inflammatory Lung Diseases

Inflammatory lung disease is a broad term for a group of diseases characterized by lung inflammation and an elevated neutrophil number. Acute and chronic inflammatory respiratory problems are both related with airway inflammation and are affected by a mix of ecological, genomic, and epigenetic factors [30].

### 8.1. Inflammation Mechanism in Lung Diseases

Inflammation is the body’s natural defensive mechanism for removing infections, toxins, and injured cells while also starting the process of healing. Pathogens or chemical exposure, allergens, contaminants, and irritants are the most common causes of airway inflammation. Toll-like receptors recognize pathogen-shared molecular processes and trigger inflammatory cells mediators such as TNF-α, chemokines, and cytokines such as interleukin 8 (IL-8) and NF-κB, which then generate growth factors to begin the corrective actions [31]. TNF-α increases the production of endothelial cell adhesion molecules in lung capillaries, while IL-8 elicits neutrophils [32]. Furthermore, several of the recognized target inflammatory proteins, including vascular cell adhesion molecule-1, cytosolic phospholipase A2, cyclooxygenase-2, intercellular adhesion molecule-1, and matrix metalloproteinase-9 (MMP-9), have been linked to airway inflammation in response to varied sensations. Severe inflammation in the lungs, which is responsible for supplying oxygen to all of the body’s organs, can be fatal. Pulmonary homeostasis requires a precise balance of inflammatory and anti-inflammatory factors. As a result, in the therapy of patients with pulmonary inflammation, a clear assessment of the inflammatory processes is critical [33,34,35]. Tissue microenvironment, stress, energy, illness, neighborhood, and seasonal variations are some of the variables that influence inflammatory reactions. In particular, contact among epithelial cells and neutrophils is possible during inflammatory reactions in which the tissue microenvironment has a direct impact on the signaling pathways and causes immune cells to rush to the damaged tissue. For many illnesses, distinct inflammatory responses have been identified, and the amplification of the inflammatory reaction that develops with age is recognized as a significant component that contributes to inflammation [36].

### 8.2. The Anti-Inflammatory Property of Vitamin D

Macrophages, an important cell type in the innate immune system, have been found to express VDR. When *Mycobacterium tuberculosis* activates the TLR1/2 heterodimer in macrophages, it causes the overexpression of VDR and CYP27B1, which culminates in the production of the antimicrobial peptide cathelicidin and the death of internalized *M. tuberculosis*. The differentiation of macrophages induced by TLR2/1 is linked to the antimicrobial pathway dependent on the vitamin D in this process, and IL-15 is involved. When the CYP27B1 levels rise, more 1,25(OH)2D3 is produced, which triggers VDR and causes target gene transcription through calcitriol responsive ingredients enshrined in the regulatory regions of the target genes of 1,25(OH)2D3 [37,38]. TLR signaling may be regulated by 1,25(OH)2D3 by modulating miR-155 in macrophages, which provides a unique negative feedback regulatory system for vitamin D to modulate the innate immune response. Vitamin D is considered to be a spontaneous endoplasmic reticulum stress reducer that can selectively inhibit IFN-γ-stimulated macrophages’ main effector activities. VDR was shown to inhibit gene transcription in the vicinity of 1,25(OH)2D3 by shifting the deoxyribonucleic acid-bound nuclear factor of stimulated T-cell, suppressing inflammatory cytokine production [39]. The most powerful antigen-presenting cells are dendritic cells (DCs). In a VDR-dependent way, calcitriol suppresses the proliferation, development and immunomodulatory ability of human DCs, characterized as tolerogenic characteristics, according to several studies. Declined costimulatory molecule surface expressions (CD86, CD80, and CD40) and major histocompatibility complex II, upregulation of inhibitory immunoglobulin-like transcript 3 molecules, and increased release of IL-10 and chemokine (C–C motif) ligand 22 are among the molecular mechanisms influencing the attenuation of tolerogenic properties of DCs by 1,25(OH)2D3 [40,41].

### 8.3. Vitamin D Dysregulation in Inflammatory Lung Diseases

Vitamin D insufficiency has been related to numerous chronic illnesses in vulnerable populations exposed to airborne particles. The bioactive version of vitamin D, calcitriol, is important for immunological modulation [42]. Many researchers have discussed the incidence and causes of calcitriol insufficiency in calcitriol, the function of vitamin D in COPD, and the vitamin D route connecting airways and inflammatory processes in COPD, among other topics [43]. Furthermore, new research has linked calcitriol deficiency in nutrition to the pathophysiology of chronic respiratory illnesses, including asthma and COPD. In vulnerable US populations residing in urban areas, the third National Health and Nutrition Examination Survey research found a significant connection between the blood levels of calcidiol and lung function, as measured by FVC and FEV1 [44]. Based on the dietary evaluation of the vitamin D status, another cross-sectional research in hospitalized adult COPD patients in Spain found similar results on reduced vitamin D consumption [45,46]. However, there was no link between the blood calcidiol levels (dietary vitamin D consumption) and lung function in spirometrically characterized COPD patients in UK population. In the general UK population, vitamin D is not a major predictor of adult lung function or COPD, according to this study. VDBP has immunomodulatory properties in the lungs and is linked to macrophage stimulation and neutrophil chemotactics [47,48]. The functional significance of the VDBP gene’s polymorphism relationship with COPD has been demonstrated in several studies. Additionally, demographic research has highlighted the connection among the reduced vitamin D blood levels and steroids used for chronic asthmatics and COPD patients, as well as individuals with a decreased glucocorticoid response. Furthermore, the impact of dietary vitamin D supplements on lung functions and lung immune–inflammatory cellular control via epigenetic chromatin changes with tobacco smoke and biological pollutants is less well-characterized [49,50].

## 9. Vitamin D-Based Therapy in Asthma

Asthma is the most common noncommunicable illness, impacting millions of people’s health. Asthma is ranked 16th globally in terms of illnesses. Asthma affects about 300 million individuals worldwide, with an additional 100 million people predicted to get the disease by 2025. It is a multigenic inflammatory disease characterized by hyper-responsiveness of the lungs. Dyspnea, wheeze, chest discomfort, and coughing are all indications of asthma. Asthma sufferers encounter a variety of signs ranging from mild to serious, which might happen daily or only rarely. Asthma attacks can occur as the symptoms grow severe. Asthma usually begins in childhood, although it may strike anybody at any age [51,52]. Adult-onset asthma refers to asthma that develops in adults, particularly women. Hereditary, race, sex, lifestyle, and medical problems are some of the risk variables for asthma. Although asthma fatality, comorbidities, and incidences are all greater in low–middle-income nations, there are substantial regional differences in asthma fatalities, comorbidities, and incidence. In asthma, several different types of cells contribute, including epithelial cells, neutrophils, eosinophils, macrophages, mast cells, and T lymphocytes [53]. Asthma is connected to T-helper cell type 2 activity. There is now no asthma-preventive approach, although treatment and action plans can help to minimize the asthma intensity and episodes. Asthma care necessitates the constant monitoring, medication, and avoidance of triggers [54]. Although there has been improvement in the asthma treatment techniques in current years, there is still room for improvement in terms of early illness detection, patient awareness and case-by-case disease prevention. The leukotrienes regulators inhaled long-acting β2-agonists, mast cell stabilizers, and corticosteroids and are now often used to alleviate inflammation and dilate airways. Even though Vitamin D performs vital functions in asthma relief, there is a paucity of tangible data due to various limitations in observational research [55]. Exogenous vitamin D therapy may assist enhance glucocorticoid responsiveness in some genetically susceptible asthma patients. NF-κB is a key controller of adaptive and innate immunity, and clinical investigations have indicated that 1,25(OH)2D2 reduces NF-κB expression, although the specific method of regulation remains unknown. Due to a lack of strong proof, the significance of Vitamin D in asthma is unknown; however, numerous cross-sectional studies have shown a link between asthma and vitamin D. Reduced concentrations of blood vitamin D have been associated with increased hospitalization and severity, as well as reduced lung performances in adolescents with asthma, according to the controlled studies [56,57]. Vitamin D supplementation has been shown in certain clinical studies to lessen the intensity of asthma. Vitamin D has an effect on pregnant women and children with asthma, according to the research. The results of the clinical research show that the supplementation of vitamin D during pregnancy is vital for preventing asthma in newborns. Increased vitamin D consumption during early pregnancy is a prenatal approach for preventing asthma, according to the researchers. Vitamin D can be utilized to lower asthma fatality and morbidity according to the research findings [58]. Vitamin D supplementation reduces the blood IL-17A levels while increasing IL-10 levels in asthma patients, according to the experimental research. IL-17 is an inflammatory cytokine that is implicated in asthma allergies and has a vital function in bacterial defense. In a mouse model of asthma, for example, vitamin D treatment suppresses airway inflammation, reduces IL4 levels, hinders T-cell recruitment, and reduces the inflammatory process (Table 1) [59].

## 10. Vitamin D Based Therapy in COPD

COPD is among the leading reasons for death globally, and it is expected to be the third-largest reason for fatality by 2030. It is classified as an inflammatory disease of the pulmonary system by the GOLD guidelines. COPD is a severe pulmonary illness marked by continuous respiration difficulties, restricted airflow, and abnormal inflammation. Breathing difficulties and inadequate airflow to the lungs are caused by inflammation in the bronchial tube linings (obstructive bronchiolitis) and breathlessness owing to alveolar sac damage (emphysema) [64,65]. The discovery of this vitamin’s numerous novel and important functions has been made possible by its deficiency. Its lack has been related to different ailments, like carcinoma, autoimmune disorders, and respiratory illnesses. Intriguingly, declined calcitriol levels have been associated with significant problems in individuals with respiratory illnesses, particularly COPD. Deficient vitamin D has been associated with the advancement of pathological conditions such as inflammatory regulation, severe oxidative stress, increased protease expression, weakened host defense, and lung airway remodeling [66]. The vitamin D (25-OH) levels in a typical adult person vary from 30 to 100 ng/mL. Deficient vitamin D (less than 20 ng/mL) in the human body can be characterized by reduced vitamin D supplementation, lack of sunshine exposure, and vitamin D malabsorption, all of which contribute to the development of COPD symptoms. Vitamin D treatment lowers the incidence of exacerbations in COPD patients who are woefully deficient in the vitamin [67]. Due to an upsurge in the synthesis of numerous matrix metalloproteinases (MMP12, MMP9, and MMP2), emphysema occurs rapidly in VDR knockout mice, resulting in inequity in the protease/antiprotease expression. Vitamin D also suppresses the TGF-b1-signaling pathway, which is linked to COPD fibrosis. Targeting oxidative stress that is important in the progression of COPD and might be a promising treatment target. Vit-D and its analogs have a possible function in activating Nrf2, according to numerous studies [68]. Vitamin D was also related to oxidative treatment. Vitamin D boosts the expression of Nrf2, which boosts alveolar macrophage phagocytosis in COPD patients. There is a growing body of data from medical studies that vitamin D administration and helps to reduce COPD exacerbation. In a double-blinded placebo-controlled randomized clinical study, for example, it was discovered that oral vitamin D treatment reduces COPD aggravation and enhances FEV1 in serious COPD patients. Another clinical trial found that vitamin D treatment enhances the respiratory performance in COPD patients with calcitriol deficiency [69]. The supplementation of vitamin D in the diet enhances COPD patients’ quality of life. At present, theophylline, long-acting beta2-agonists (LABA), and inhaled corticosteroids (ICS) are the most often utilized medications to treat individuals with COPD. The major function mechanisms of these medications are to eradicate inflammation and to relax the bronchial tube to lower the airway resistance. Interestingly, vitamin D sufficiency is probably advantageous in guarding against airway and systemic inflammatory illnesses. Many researchers have explored that vitamin D administration can enhance COPD symptoms. According to earlier research, vitamin D might raise the blood levels of the previously listed medications and improve patients’ lung function [70].

Thus, vitamin D treatment in COPD patients with deficient vitamin D may be a useful and effective therapeutic strategy to avoid the disease from deteriorating. COPD exacerbation is connected to a high incidence of deficient vitamin D, although low vitamin D blood concentrations are not linked to susceptibility [71]. Vitamin D supplementation can help avoid COPD aggravation. As a result, an additional study is aimed at understanding the significance of vitamin D in COPD patients, as definitive data from many interventional trials are lacking (Table 2).

## 11. Vitamin D Based Therapy in Lung Cancer

Lung cancer is the commonest reason of carcinoma-related death in both sexes across the globe. Non-small-cell lung cancer (NSCLC), which contributes to 85% of all cases of pulmonary carcinoma, and small-cell lung cancer (SCLC), which accounts for 15% of the cases, are the two primary types of lung cancer. The carcinoma of squamous cells, adenocarcinoma, and lung carcinoma of the large cells are the three types of NSCLC [75,76]. While smoking tobacco is the leading cause of lung cancer, an approximated 25% of instances occur among nonsmokers, most commonly in the form of adenocarcinomas. Vitamin D has been widely investigated in a variety of cancer situations, and there is compelling proof that it is antineoplastic [77].

Vitamin D’s anticancer properties are considered to be triggered by calcitriol interacting with the VDR. Lung cancer cell growth is inhibited, apoptosis is promoted, and angiogenesis is reduced through these methods. A recent in vitro study found that calcitriol caused G0/G1 cell cycle blockage by downregulating cyclins that facilitate cell cycle entrance into the S phase. 1,25(OH)2D inhibits angiogenesis by inhibiting the production of VEGF, which is reported to stimulate endothelial cell excitation, recruitment, and multiplication. In lung cancer cells treated with 1,25(OH)2D, parathyroid hormone-related protein (PTHrP), MMP-9, and MMP-2 expression and synthesis are similarly decreased [78,79]. Since PTHrP and MMPs are both important factors in carcinoma progression, this might be a major mechanism. In a mouse model of lung cancer, the researchers discovered that supplementing with 1,25(OH)2D reduced the tumor occurrence and dramatically reduced tumor multiplication in a dose-dependent way [80]. In vivo, 1,25(OH)2D can reduce pulmonary carcinoma cell metastasis development, implying that sufficient vitamin D levels can inhibit pulmonary carcinoma etiology. While the research to date shows that the vitamin D level influences the start of tumor development, additional clinical research is required to establish if vitamin D can inhibit tumorigenesis (Table 3) [81].

## 12. Vitamin D Based Therapy in Pulmonary and Cystic Fibrosis

Idiopathic pulmonary fibrosis (IPF) or pulmonary fibrosis is a long-term lung fibrotic infection characterized by a mix of hereditary and ecological variables. IPF can also be caused by cystic fibrosis. Cystic fibrosis (CF) is a fatal autosomal-recessive genetic condition that shortens the lifespan. It mostly affects the lungs, pancreas, and other organs, resulting in blockage and inflammation. A polymorphism in the CF transmembrane conductance regulator (CFTR) gene causes it [81]. The FDA-approved medications for IPF include pirfenidone and nintedanib; however, neither of these drugs is very successful. Furthermore, vitamin D insufficiency has been associated with pulmonary fibrosis and the deterioration of lung function, although the basic concept is unclear. According to studies, vitamin D insufficiency affects 90% of CF patients. Vitamin D deficiency is common in CF patients with pancreatic insufficiency owing to fat malabsorption. Vitamin D supplements are commonly given to children with CF at a young age; however, too much vitamin D might cause respiratory difficulties [86]. Supplemental vitamin D doses and a routine review of the vitamin concentrations must be performed for people to maintain adequate levels. Lung transplantation, which demands an appropriate amount of vitamin D, can enhance the comfort and duration of life in certain CF patients. Concurrently, it was discovered that vitamin D reduces the occurrence of transplant rejection. Vitamin D is unquestionably important along the course of CF [87]. It is worth noting that vitamin D has a positive physiological action on the morbidities and the intricacy that comes with this fundamental condition. As a result, more research is needed to confirm the undisputed potential for alleviating fibrosis complications. Vitamin D supplementation was advised in CF patients with *P. aeruginosa* infection in a scientific investigation that had anti-inflammatory properties by lowering the level of IL-23 and IL-17A [88]. Vitamin D therapies were tested on paraquat (PQ)-induced paralysis. Male C57/BL6 mice with lung fibrosis had fewer leukocytes in their BALF and lower levels of MMP-9, TGF-β, IL-17, and IL-6. Vitamin D treatment in mice may reduce bleomycin-induced lung fibrosis, according to the research. Vitamin D combined with DNA-damaging chemicals might be utilized to treat pulmonary fibrosis. RAS can be triggered by a lack of vitamin D [89]. RAS overexpression has been linked to pulmonary fibrosis. A study found that blood samples from CF patients with high 25(OH)D levels are strongly linked to pulmonary function, although further research is needed to confirm this. According to another study, increasing the vitamin D blood levels can reduce respiratory complications [90]. Ultimately, there are some signs that vitamin D may be a viable therapeutic strategy for CF and IPF, but further large-scale research is needed. Personalized therapy and pharmacogenomics research has some validity for illness therapy on a case-by-case basis (Table 4).

## 13. Vitamin D Based Therapy in Pulmonary Infection including COVID-19

### 13.1. Pneumonia

Pneumonia is a serious pulmonary illness that has a high fatality rate. It leads to pus and fluid in filled alveoli, resulting in inflammation of the alveoli and adjacent tissue, resulting in a sluggish inhalation of oxygen. Indications such as sputum cough, high temperature, muscular tiredness, breathlessness, and others are seen as consequences. Bacteria (such as *Pseudomonas aeruginosa, Haemophilus influenzae, Streptococcus pneumonia,* and *Staphylococcus aureus*); viruses (such as Coronaviruses, Respiratory Syncytial Virus, Influenza A and B, and others); fungi; and parasites can all cause pneumonia [94,95]. As per the World Health Organization, it is the leading cause of mortality among children globally, with an estimated 1.4 million child fatalities each year. In 2017, there were over 808,694 child fatalities confirmed, accounting for 15% of all child deaths under the age of five. Calcitriol has been associated with pneumonia in several studies. Inadequate vitamin D levels can cause pneumonia in newborns and hospitalization in individuals with serious instances of pneumonia. A decreased amount of calcidiol in the blood of the placental cord has been related to an increased risk of pneumonia in infancy [96,97]. As a result, we may deduce that vitamin D insufficiency in mothers may lead to vitamin D deficit in children; therefore, women should consume enough amounts of vitamin D throughout pregnancy, either through nutrition or supplementation. Pneumonia is typically classified as either hospital-acquired or community-acquired, but individuals with vitamin D deficiency are more likely to develop severe community-acquired pneumonia (CAP), resulting in prolonged stays in clinics and critical care units. Patients admitted to the hospital with vitamin D deficiency and acute ischemic stroke are more prone to developing stroke-associated pneumonia [98]. As the vitamin D level falls, the risk of pneumonia rises. Many researchers have found a link between pneumonia and vitamin D deficiency; however, these investigations yielded ambiguous results.

### 13.2. Tuberculosis

After HIV/AIDS, tuberculosis (TB) is the second-most well-known infectious illness, accounting for more than 2 million fatalities worldwide each year. A mild respiratory infection occurs during the early stages of tuberculosis, and a failure in identification and management at this point allows the infection to spread easily through sneezing and coughing [99]. Regardless of the reality that just 10% of the global population with latent tuberculosis will manifest as the aggressive form of the illness, determining which individuals will advance through the disease and will preserve their immunological control or resolve it becomes a critical question with significant health implications [100]. People with tuberculosis are frequently impoverished, which causes the immune response to deteriorating. As a result, dietary supplements that include both micro- and macronutrients may be helpful. Vitamins have long-been thought to be important immune enhancers. Vitamin D has been found to have antimycobacterial effects in recent research. The level of vitamin D is among the extrusive risk deciding variables. Studies on high-risk individuals have looked at several host factors that lead to comprehensive knowledge of the active TB pathway. Periodic variations that cause a decrease in type B UV radiation, lack of sunshine exposure, and dietary insufficiency have all been linked to vitamin D deficiency. Two recent studies have confirmed a substantial link between the seasonal changes in vitamin D blood levels and the prevalence of tuberculosis. A meta-analysis research found a link between poor vitamin D levels and an elevated chance of tuberculosis. Vitamin D supplementation, as a result, might be a powerful tool for reducing the risk of tuberculosis, maintaining the immunity of the host, and improving the antituberculosus therapy efficacy, and it is a hot topic in the current study [101]. Vitamin D’s immunostimulatory properties (which include the activation of innate antibacterial activity or actions, as well as anti-inflammatory pathways) have also helped it gain widespread interest. Infection with Mycobacterium tuberculosis causes cellular injury by increasing the production of MMP-10, MMP-1, and MMP-7 in macrophages. MMP-9 has been linked to the intensity of the tuberculosis and the development of TB granulomas [102]. MMP-9 has been related to tuberculosis severity and the formation of tuberculoma granulomas. In epithelial cells infected with Mtb, tissue regulators of MMPs (TIMP) were shown to be reduced. Vitamin D therapy of PBMCs decreases the MMP-10, MMP-1, and MMP-7 expression while increasing the TIMP-1 expression, indicating its critical function in preventing cellular injury and providing symptom relief while having an infection [103]. Additionally, vitamin D deficiency enhances patients’ vulnerability to tuberculosis, as well as the likelihood of the illness progressing from dormant to aggressive, supporting its use as a prophylactic in patients with latent TB [104].

### 13.3. COVID-19

The new coronavirus illness (COVID-19), which was first discovered in late 2019 in Wuhan, China, is still spreading throughout the world, infecting more than 100 million people and killing approximately 2.4 million people in 221 nations around the globe. Due to the pandemic’s fast spread and very substantial fatality rates, particularly among vulnerable groups, controlling and preventing it has proven difficult [105,106]. The infection has now quickly spread to nearly every part of the globe, claiming many lives, hurting economies, and jeopardizing recent medical achievements. Several cellular mechanisms have been recognized in associated to a virus of COVID-19, including RIG-I and MDA5 host–recognition evasion, the disruption of M–protein-mediated type-1 IFN induction, dipeptidyl peptidase-4 receptor binding, and papain-like protease-mediated replication [107]. Human DPP-4/CD26 is manifested to interplay with the S1 domain of the COVID-19 spike glycoprotein, representing that it might be a vital virulence factor in COVID-19 infection. When the vitamin D deficiency is corrected, the DPP-4/CD26 receptor expression is dramatically decreased in vivo [108]. There is also proof that managing calcitriol levels may lessen some of the unfavorable downwind immunological complications associated with the infection of COVID-19, such as an IL-6 surge, postponed IF-γ reactions, and an adverse prognostic indicator in individuals with acute illness pneumonia, even those with COVID-19 infectious disease [9,109]. Recent studies have shown some of the mechanisms through which calcitriol reduces the incidences of viral infections (Figure 3).

Vitamin D minimizes the incidences of viral infection and death in a variety of ways. Vitamin D works through three mechanisms to lower the chance of catching a cold: cellular natural immunity, adaptive immunity, and the physical barrier [110]. Many in vitro investigations have shown that vitamin D performs an important function in local “respiratory homeostasis”, either by promoting the production of antiviral proteins or by interacting with pulmonary virus multiplication. Some retrospective investigations revealed a link between COVID-19 and vitamin D instances and outcomes, whereas others found no link when influencing the factors that were taken into account [59,111]. There is, however, the inadequate information to link vitamin D levels to COVID-19 intensity and death (Table 5).

## 14. Excessive Use of Vitamin D Pharmaceutical Formulations

Vitamin D is a necessary prohormone that is required for the maintenance of healthy bones and calcium levels. Vitamin D insufficiency results in hypocalcemia and bone mineralization abnormalities. Vitamin D has become a common alternative due to the rising awareness of vitamin D insufficiency and associated health issues, and its use has expanded significantly. Increased vitamin D supplementation by the general population and an increasing number of therapeutic dosage prescriptions (including extremely high doses) without medical supervision may result in an increased risk of exogenous hypervitaminosis D, commonly known as vitamin D toxicity [115].

Hypervitaminosis D, along with hypercalcemia, occurs as a result of uncontrolled usage of megadoses of vitamin D or vitamin D metabolites. Hypervitaminosis D may develop in some clinical circumstances as a result of the use of vitamin D analogs (exogenous vitamin D toxicity). Hypervitaminosis D, in conjunction with hypercalcemia, may also be a symptom of excessive 1,25(OH)2D production in granulomatous diseases, lymphomas, and idiopathic infantile hypercalcemia (IIH). Exogenous vitamin D toxicity is often induced by the chronic usage of megadoses of vitamin D, not by excessive sun exposure or a diverse diet. The human body can control the amount of previtamin D (tachysterol and lumisterol) generated by ultraviolet B light in the skin. The vitamin D levels in a varied diet are often low, and the vitamin D fortification of food items is minimal [116]. Exogenous vitamin D toxicity is defined by very increased 25(OH)D levels (>150 ng/mL), severe hypercalcemia and hypercalciuria, and extremely low or undetectable parathyroid hormone (PTH) activity. The earliest quantifiable signs of vitamin D poisoning are hypercalciuria and hypercalcemia. In individuals with vitamin D toxicity, the 1,25(OH)2D concentration may be within the reference range, slightly raised, or decreased (less commonly) when a high calcium level in the serum reduces the PTH activity. 1,25(OH)2D is suppressed by inhibiting 1-hydroxylase activity and increasing the 24-hydroxylase activity [117].

## 15. Conclusions

Vitamin D is an essential vitamin that influences the physiological activities, such as cellular differentiation and proliferation, host defense, immunological regulation, and inflammation, in addition to the usual bone and calcium homeostasis functions. Deficient vitamin D has been associated with different problems related to health, such as bone softening and skeletal abnormalities, as well as immunological disorders. A vitamin D deficit is also linked to inflammatory lung diseases, such as COVID-19, asthma, lung cancer, COPD, pulmonary and cystic fibrosis, pneumonia, and TB putting people with a vitamin D deficiency at a greater chance of developing respiratory diseases with substantial morbidity and fatality. As a result, the interrelationships among vitamin D and respiratory illnesses have sparked a new age of attention in supplementing as a cost-effective way to enhance world health. Furthermore, the clinical usage of vitamin D would aid in immunological sustenance, which would be an exciting future prospect.

## Figures and Tables

**Figure 1 biomedicines-09-01843-f001:**
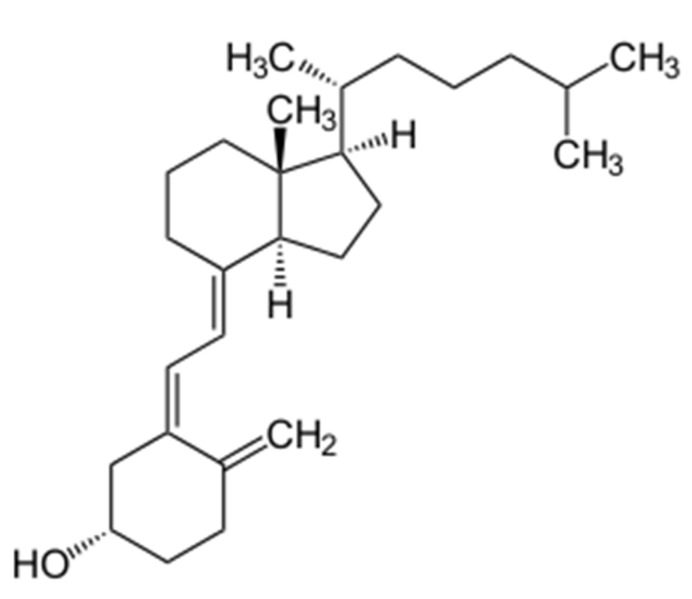
Structure of Vitamin D_3_.

**Figure 2 biomedicines-09-01843-f002:**
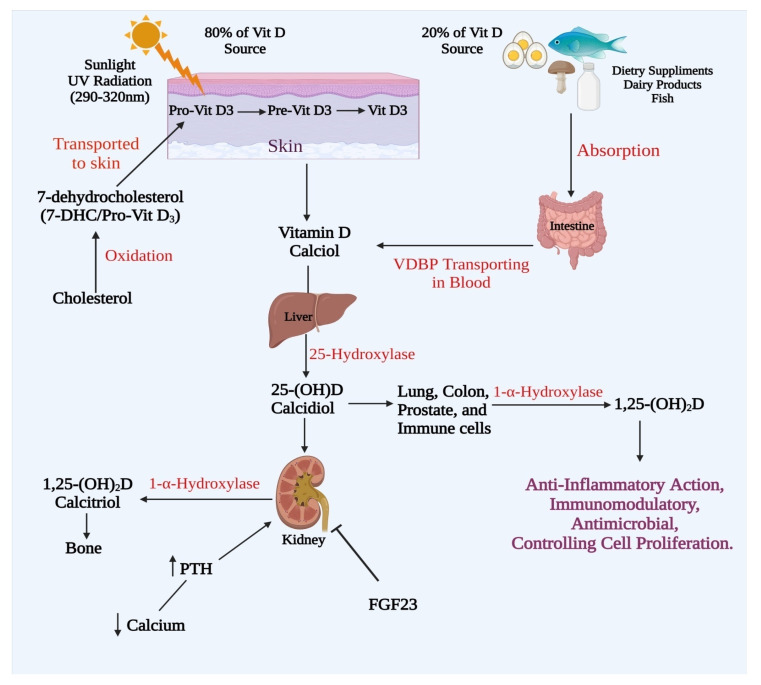
Pathway of calcitriol metabolism.

**Figure 3 biomedicines-09-01843-f003:**
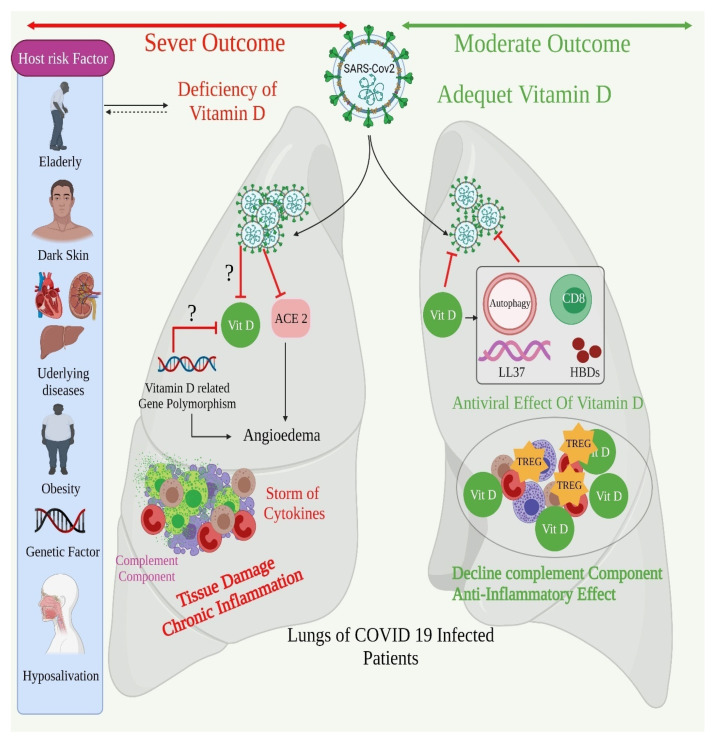
Mechanism of vitamin D in combating COVID-19.

**Table 1 biomedicines-09-01843-t001:** Studies on the investigation of vitamin D role in combating asthma.

Study Type	Study Design	Parameter Examined	Findings	Ref
Clinical Trial	At 3 US centers, Placebo controlled randomized double-blind trial in woman, Placebo (*n* = 436) *v*/*s* 4000 IU/d vitamin D (*n* = 447), administered parentally.	Through the age of three, parents reported having asthma or recurrent wheeze, according to a physician’s diagnosis; Level of calcitriol in pregnant woman 3rd trimester.	Supplementing with 4400 IU/d vitamin D vs. 400 IU/d vitamin D substantially enhanced vitamin D levels in pregnant mothers at risk of getting an asthmatic child. At age three, their children had a 6.1 percent declined asthma incidence and recurrent wheeze, although this did not reach statistical significance.	[60]
	Placebo-controlled, randomized double-blind trial, Physician diagnosed children between ages 5–13 years with moderate to severe asthma, Vitamin D_3_ 60,000 IU/month for *v*/*s* Placebo.	Asthma exacerbation	At 6 months, the D3 treated group had a significantly higher reduction in asthma severity as per GINA standards.	[61]
Cross-sectional Studies	5110 Physician diagnosed asthma patients of age between 50–84 years treated with vitamin D supplement.	Asthma exacerbation.	Asthmatic individuals with vitamin d Deficiency are more inclined to seek emergency medical attention for their asthma and have poor asthma management.	[62]
Preclinical Studies	BALB/c mice	Measurement of in vivo AHR; cytokine production and proliferative reactions to OVA;inflammatory cells.	Suppresses AHR and airway cellular response; Reduces the severity of asthma by reducing mediator release.	[63]
	Sprague–Dawley rats; Vitamin D 100 ng/mL	Airway remodeling; Asthma exacerbation.	Vitamin D treatment reduced airway remodeling in asthma patients by inhibiting the Wnt/β-catenin signaling pathway.	[57]

**Table 2 biomedicines-09-01843-t002:** Studies on the investigation of the role of vitamin D in combating COPD.

Study Type	Study Design	Parameter Examined	Findings	Ref
Clinical Studies	Multi-center, randomized, double-blind, placebo-controlled intervention trial, age > 40 years, 16,800 IU vitamin D3 (*n* = 120) *v*/*s* placebo (*n* = 120) weekly and orally.	COPD exacerbation, Total lung capacity, and maximum respiratory mouth pressure.		[72]
	Randomized clinical study with a double-blind placebo control, 88 severe COPD patients, placebo receive 100,000 IU vitamin D monthly for six months.	FEV1, COPD exacerbation.	Improved FEV1, Reduces COPD exacerbation.	[40]
	Controlled, randomized, double-blind trial, 50–58 year patents, 200,000 IU followed by 100,000 IU vitamin D monthly for 1.1 years (*n* = 226) *v*/*s* placebo (*n* = 216).	FEV1, COPD exacerbation.	Only smokers benefitted from vitamin D supplementation, particularly those with vitamin D insufficiency or COPD.	[73]
	Multi-center, randomized, double-blind, placebo-controlled intervention trial, Vitamin D_3_ (*n* = 122) *v*/*s* Placebo (*n* = 118)	COPD exacerbation.	Vitamin D_3_ supplementation reduced the severity of COPD exacerbations in those with mild to severe COPD.	[74]

**Table 3 biomedicines-09-01843-t003:** Studies on the investigation of the old vitamin D role in combating lung cancer.

Study Type	Study Design	Parameter Examined	Findings	Ref
Clinical Studies	Double-blind, randomized trial, Vitamin D 1200 IU/d (*n* = 77) *v*/*s* placebo (*n* = 78)	Overall survival and relapse-free survival.	Patients with early-stage lung adenocarcinoma may benefit from vitamin D therapy.	[82]
Preclinical Studies	A/J Mouse model, Vitamin D_3_ (2.5 or 5 microgram/Kg diet)	Tumor incidence and tumor cell differentiation.	Reduces incidence of the tumor as well as having combating potential against lung carcinogenesis.	[83]
	Mouse model of N-nitroso-tris-chloroethyl urea; Vitamin D_3_ 2000 IU/Kg.	The premalignant tumors progressing of Carcinoma	Reduces proliferation, development of premalignant lesion, swelling of squamous cell carcinoma of the lung.	[84]
In vitro studies	NCI-H1975 and A549 tumor cells	Metastasis, tumor cell apoptosis.	The tumor cell growth, infiltration, and metastasis are inhibited, while tumor cell apoptosis is promoted.	[85]

**Table 4 biomedicines-09-01843-t004:** Studies on the investigation of the vitamin D role in combating pulmonary and cystic fibrosis.

Study Type	Study Design	Parameter Examined	Findings	Ref
Clinical Studies	Randomized open-labeled intervention, 16 Cystic fibrosis patients receive Vitamin D_3_ 35,000 IU/week for age < 16 years or 50,000 IU/week for age > 16 years for 3 months	T cell activation, myeloid dendritic cells.	In people with CF, vitamin D has a wide range of immunomodulatory effects	[91]
	Multicenter, randomized, double-blind, placebo-controlled intervention trial, 23 CF patients chronically affected withP. aeruginosa receive 1000 IU/d for 3 months *v*/*s* Placebo orally.	Quantification of IL-17A and IL-23.	Vitamin D had an anti-inflammatory impact, lowering the levels of IL-17A and IL-23 in CF patients’ airways. Vitamin D supplementation is recommended for CF patients.	[17]
Preclinical Studies	C57/BL6 male mice, Vitamin D I.P. daily at a dose of 5 μg/kg.	Leucocyte count, estimation of inflammatory mediators.	Vitamin D decreases leucocyte count; reduces the level of MMP-9, TGF-β IL-17, and IL-6; beneficial effect in PF treatment	[19]
	C57/BL6 mice treated with bleomycin, Vitamin D 1 μg/kg/day between 3rd day–13th days.	Level of hydroxyproline, Masson Trichrome staining and level of mRNA α-SMA, col3a1 and col1a1.	Up-regulation of mRNA of VDR level, Vitamin D hasthe potential of combating IPF.	[92]
In vitro Studies	Human myofibroblasts, Alveolar epithelial cells type II	DNA damaging	In the vicinity of a DNA damaging chemical in PF, vitamin D had an unexpectedly negative effect.	[93]

**Table 5 biomedicines-09-01843-t005:** Studies on the investigation of the vitamin D role in combating COVID-19.

Study Type	Study Design	Parameter Examined	Findings	Ref
Clinical Trials	Multi center, open-label, randomized controlled trial, Vitamin D 50,000 IU daily orally to 260 COVID19 Patients of age ≥ 65 years.	All Causes of mortality.	Vitamin D supplementation at high doses might be an efficient, well-tolerated, and quickly available therapy for COVID-19.	[112]
	Household cluster-randomized with a planned pragmatic, double-blinded trial, 2700 subjects 1:1 ratio vitamin D 3200 IU/d *v*/*s* Placebo.	The likelihood of hospitalization and/or fatality among newly diagnosed people.	Lowering hospitalization and/or death rates in recently diagnosed patients, as well as avoiding infection within their intimate infected persons	[113]
	Open-label randomized parallel pilot, double-blinded trial, 76 COVID-19 hospitalized patients.	ICU admissions and fatalities rate	The use of calcifediol has been shown to minimize the requirement for ICU care in individuals who require hospitalization. COVID-19	[109]
	Randomized Multicenter clinical trials, 69 COVID-19-positive patients, 5000 IU/d (*n* = 36); 1000 IU/d (*n* = 33) for two weeks orally.	Gustatory sensory loss and cough recovery	The time it takes for patients to recover from gustatory sensory loss and cough is reduced by taking 5000 IU of vitamin D3 daily for two weeks.	[114]
	65 hospitalized COVID-19 positive patients of age between 63–89 years.	Commodities, Type of respiratory involvement, laboratory parameters (vitamin, C-reactive protein, D, D-dimer), Pulmonary parameters (PaO_2_/FiO_2,_ PaCO_2_, PaO_2_, and SO_2_)	Vitamin D insufficiency is linked to more serious respiratory involvement, a lengthier illness period, and a higher chance of mortality.	[112]

## Data Availability

Not applicable.

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
