# Peer review of "Current Overview on Therapeutic Potential of Vitamin D in Inflammatory Lung Diseases"

_biomedicines, 2021, doi:10.3390/biomedicines9121843_

Round 1

Reviewer 1 Report

Dear Authors I send you my comments:

1) Methods: please add the method used for manuscript evaluation

2) Results: Please add if there is a differetiation for gender concerning the time, the dosage and the efficacy of vitamin d. 

References:

line 58-66 please add references

Line 167-173 add references (see and cite reference: doi 10.1002/jcb.20124)

Line 181 please add reference : doi: 10.1155/2015/879783

Line 214-223 please add reference

Line 460 add reference: doi: 10.1002/jcph.1644.

Author Response

Dear Authors I send you my comments:

  • Methods: please add the method used for manuscript evaluation

Response: Dear reviewer, I have included a separate section of method of literature search after introduction.

  • Results: Please add if there is a differetiation for gender concerning the time, the dosage and the efficacy of vitamin d. 

Response: Dear reviewer, I have included a separate section as a “Vitamin D consumption and status” in revised manuscript.

References:

line 58-66 please add references

Line 167-173 add references (see and cite reference: doi 10.1002/jcb.20124)

Line 181 please add reference :doi: 10.1155/2015/879783

Line 214-223 please add reference

Line 460 add reference: doi: 10.1002/jcph.1644.

Response: Dear reviewer, as per your comments I have included a references.

Reviewer 2 Report

In this review, the authors focus on how Vitamin D moderates the immune system and the benefit of applying Vitamin D to lung disease. Therefore, with moderate reference research, the authors outline the therapeutic potential of Vitamin D. However, the authors need to more detail depict the combination of other medicinal drugs during the COPD treatment since Vitamin D is a nutritional molecular. Also, the authors need to discuss the drawback when taking too much Vitamin D.

drugs during the COPD treatment since Vitamin D is a nutritional molecular. Also, the authors need to discuss the drawback when taking too much Vitamin D.

Minor concern

Authors need to check the spelling and grama used carefully. For example, in lines 59-61, “The phosphodiesterase-4 inhibitors, methylxanthins, orally as well as inhaled corticosteroid, and monoclonal antibodies are the anti-inflammatory agents used in management of ILDs.” need to be modified as “The phosphodiesterase-4 inhibitors, methylxanthines, orally as well as an inhaled corticosteroid, and monoclonal antibodies are the anti-inflammatory agents used in the management of ILDs.”

Author Response

In this review, the authors focus on how Vitamin D moderates the immune system and the benefit of applying Vitamin D to lung disease. Therefore, with moderate reference research, the authors outline the therapeutic potential of Vitamin D. However, the authors need to more detail depict the combination of other medicinal drugs during the COPD treatment since Vitamin D is a nutritional molecular. Also, the authors need to discuss the drawback when taking too much Vitamin D.

Response: Dear reviewer, as per your comments, we have included a paragraph about  vitamin d role in presence of other drug administration. Also included a section “Excessive use of vitamin D pharmaceutical formulations”.

Minor concern

Authors need to check the spelling and grama used carefully. For example, in lines 59-61, “The phosphodiesterase-4 inhibitors, methylxanthins, orally as well as inhaled corticosteroid, and monoclonal antibodies are the anti-inflammatory agents used in management of ILDs.” need to be modified as “The phosphodiesterase-4 inhibitors, methylxanthines, orally as well as an inhaled corticosteroid, and monoclonal antibodies are the anti-inflammatory agents used in the management of ILDs.”

Response: Dear reviewer, we have done a correction as per your comment. Also corrected grammatical mistakes in whole manuscript.

Round 2

Reviewer 1 Report

Dear Authors,

I have read the manuscrpt and I have not further comments